# Evolution of empathetic moral evaluation

**Arunas L Radzvilavicius[1]\*, Alexander J Stewart[2], Joshua B Plotkin[1]\***

[1]Department of Biology, University of Pennsylvania, Philadelphia, United States;
[2]Department of Biology, University of Houston, Houston, United States

**Abstract** Social norms can promote cooperation by assigning reputations to individuals based on their past actions. A good reputation indicates that an individual is likely to reciprocate. A large body of research has established norms of moral assessment that promote cooperation, assuming reputations are objective. But without a centralized institution to provide objective evaluation, opinions about an individual's reputation may differ across a population. In this setting we study the role of empathy–the capacity to form moral evaluations from another person's perspective. We show that empathy tends to foster cooperation by reducing the rate of unjustified defection. The norms of moral evaluation previously considered most socially beneficial depend on high levels of empathy, whereas different norms maximize social welfare in populations incapable of empathy. Finally, we show that empathy itself can evolve through social contagion. We conclude that a capacity for empathy is a key component for sustaining cooperation in societies.
DOI: https://doi.org/10.7554/eLife.44269.001

## Introduction

Widespread cooperation among unrelated individuals in human societies is puzzling, given strong incentives for exploitative cheating in well-mixed populations (*Ohtsuki et al., 2006*). Theories of cooperation based on kin selection, multilevel selection, and reciprocal altruism (*Nowak, 2006*) provide some insight into the forces driving prosocial behavior, but in human societies cultural forces appear to be of much greater importance (*Gintis et al., 2003*; *Buckholtz and Marois, 2012*). One possible explanation rooted in cultural norms is that humans condition their behavior on moral reputations: the decision to cooperate depends on the reputation of the recipient, which itself depends on the recipient's previous actions (*Leimar and Hammerstein, 2001*; *Nowak and Sigmund, 2005*). Altruistic behavior, for instance, may improve an individual's reputation and confer the image of a valuable member of society, which attracts cooperation from others in future interactions (*Nowak and Sigmund, 1998*).

Game theory has been used to study how reputations might facilitate cooperation in a population engaged in repeated social interactions, such as the Prisoner's Dilemma or the Donation Game (*Rapoport et al., 1965*; *Nowak and Sigmund, 2005*). In the simplest analysis an individual's reputation is binary, either 'good' or 'bad', and the strategy of a potential donor depends on the recipient's reputation (*Ohtsuki and Iwasa, 2004*) – for example, cooperate with a good recipient and defect against a bad recipient. A third-party observer then updates the reputation of the donor in response to her action towards a recipient. Reputation updates are governed by a set of rules, known as a *social norm*, which prescribe how an individual's reputation depends on her actions during social interactions.

A common simplification in models of moral reputations is that all reputations are both publicly known and fully objective (e.g. *Nowak and Sigmund, 1998*; *Pacheco et al., 2006*; *Ohtsuki et al., 2009*; *Sasaki et al., 2017*). This means that all individuals know the reputations of all members of the society, and personal opinions about each individual's reputation do not differ. This is a reasonable assumption if there is a central institution that provides objective moral evaluation, or if opinions regarding reputations homogenize rapidly through gossip (*Nowak and Sigmund, 2005*). But

\*For correspondence:
arunas@sas.upenn.edu (ALR);
jplotkin@sas.upenn.edu (JBP)

**Competing interests:** The authors declare that no competing interests exist.

**eLife digest** When meerkats have pups, they employ an individual to stand guard and warn the others of potential dangers and predators, putting their own life at risk. What seems like a selfless act is actually a common behavior found throughout the animal kingdom. But rather than acting out of concern for another ones wellbeing, it is considered to be an altruistic behavior towards kin, where an individual sacrifices its own reproductive success for the sake of the reproductive fitness of its entire clan.

In human societies, however, people often act altruistically towards unrelated individuals and have developed sophisticated systems of moral evaluation to decide who is worthy of cooperation and likely to reciprocate a favor. In other words, individuals will only help those who have a good reputation for being altruistic themselves. However, for this system to work, reputations need to be public knowledge, and societies need to agree on everyones reputations. But what happens when opinions about an individual's reputation are private and vary across a population?

Now, Radzvilavicius et al. wanted to find out whether altruism can emerge when people have different opinions about each others moral reputations. To do so, they used a so-called evolutionary game theory a mathematical description of how strategies change in a population over time. In their model, each individual could decide if they wanted to pay a personal cost to create a benefit for another individual. Each participant decided whether to act altruistically based on the reputation of the recipient; observers could update the individuals reputation based on their behavior.

The mathematical model revealed that when people are more empathetic and able to put themselves in someone elses shoes, altruism tends to spread over time. When people take into account different opinions and form moral judgements from another person's perspective, the population can sustain a higher level of cooperation. Moreover, the capacity for taking another person's perspective can itself evolve and remain stable in a population meaning that those individuals who evaluate each other empathetically tend to do better, and empathy spreads through social influence.

These findings can help us understand how empathy might have evolved in societies that value reputation as a means of reciprocity. A next step could be to test the theory developed by Radzvilavicius et al. in manipulative experiments, or to compare the theory to field data on reputations and behavior in online interactions.

DOI: https://doi.org/10.7554/eLife.44269.002

these conditions are rare in human populations, and opinions about reputations typically differ among individuals – for instance, because observers use different moral evaluation rules, or because of divergent observation histories, or errors. In these cases a single focal individual may have different reputations in the eyes of distinct observers, resulting in much lower rates of sustained cooperation (*Okada et al., 2017*; *Hilbe et al., 2018*).

Moral relativity – that is, when an individual's reputation depends on the observer – introduces an interesting and overlooked ambiguity in how an observer should evaluate a donor interacting with a recipient. One approach is to assume that the observer can refer only to her own opinion of the recipient's reputation, when evaluating a donor. We call this an 'egocentric' judgment, because the observer makes moral evaluations solely from her own perspective (*Figure 1a*). Alternatively, an observer can perform a moral evaluation that accounts for the recipient's reputation in the eyes of the donor (*Figure 1b*). This 'empathetic' case requires that the observer take the perspective of another person, which assumes some capacity for recognizing the relativity of moral status.

Psychological studies implicate empathy as potent driver of prosocial and cooperative behavior in human societies (*Eisenberg and Fabes, 1990*; *Batson et al., 1997*; *Batson and Moran, 1999*; *Decety et al., 2016*). The cognitive capacity to intentionally adopt the subjective perspective of another individual is known as a key component of empathetic behavior (*Davis, 1983*). This so-called 'perspective-taking' component of empathy is in turn related to the theory of mind, or the ability to attribute mental states to explain and predict the behavior and emotions of other individuals (*Premack and Woodruff, 1978*; *Hughes and Dunn, 1998*). Empathetic perspective-taking generally develops between infancy and pre-school years, with at least some components learned from

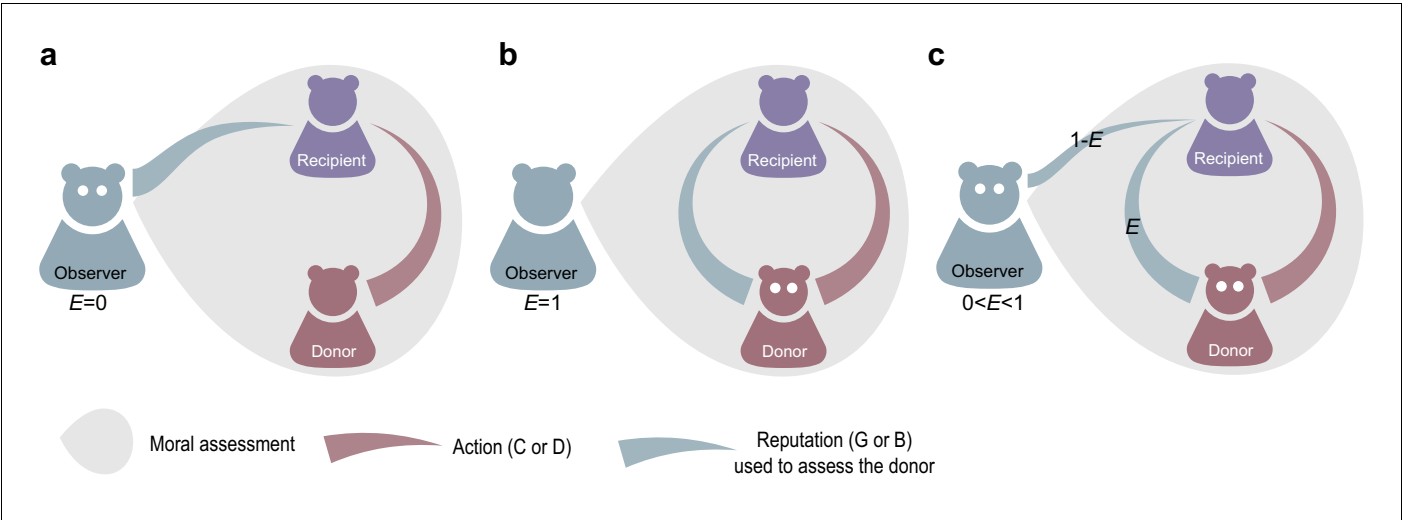

**Figure 1.** Empathetic and egocentric modes of moral assessment. An observer updates the reputation of a donor based on the donor's action towards a recipient and the recipient's reputation. (a) An egocentric observer ($E = 0$) forms a moral judgment based on the recipient's reputation as seen from her own perspective. (b) An empathetic observer makes a judgment based on the recipient's reputation in the eyes of the donor ($E = 1$). (c) More generally the empathy parameter $E$ corresponds to the probability that observer will assess the donor using the donor's – not the observer's – perspective of the recipient's reputation.

DOI: https://doi.org/10.7554/eLife.44269.003

parents (*Krevans and Gibbs, 1996*; *Knafo et al., 2008*; *Farrant et al., 2012*). And yet empathetic behavior is not universal, as even adults often fail to empathize, especially in interactions with unfamiliar social or different cultural groups (*Cikara et al., 2014*). In the context of social dilemmas, it has been suggested that empathy might play a role in evaluating the 'fairness' of opponents' actions and predicting their strategies (*Singer and Fehr, 2005*). However, the role of empathy for moral evaluation of social behavior has not been thoroughly studied. In particular, there is currently no formal way to analyze the role of empathetic perspective-taking in game-theoretic models of human cooperation.

Here we work to resolve the ambiguity of subjective moral judgment by introducing the concept of empathy into game-theoretic analyses of cooperation. We treat empathy $E$ as the probability that an observer will form moral evaluations from the perspective of another person (*Figure 1c*). First we investigate the effects of empathy on the level of sustained cooperation under simple social norms, while players update their strategies. Next we consider evolution of empathy itself using the formalism of adaptive dynamics; and we determine conditions under which empathy will evolve and remain evolutionarily stable.

## Model

### A model of moral assessment

We consider a population of individuals who can choose between cooperation or defection in a sequence of pairwise, one-shot donation games. In a given game the donor must choose whether or not to cooperate with the other player. If a donor cooperates she pays the cost of an altruistic act $c$, while the recipient receives the benefit $b > c$; if the donor defects she incurs no cost, and the recipient receives no benefit. The donation game is therefore a special case of the prisoner's dilemma (*Rapoport et al., 1965*) characterized by the payoff matrix $\begin{pmatrix} b - c & -c \\ b & 0 \end{pmatrix}$.

The decision to cooperate or defect depends on the donor's strategy $S = (p, q)$, which prescribes an action conditioned on the reputation of the recipient. Here $p$ and $q$ denote the probability that the donor will cooperate with a 'bad' or a 'good' recipient, respectively. As is common in the game-theoretic literature with reputations (*Sasaki et al., 2017*), we assume that both $p$ and $q$ are in $\{0, 1\}$,

and so we focus on three strategies: Always Cooperate ALLC, $S = (1,1)$); Always Defect (ALLD, $S = (0,0)$); and Discriminate (DISC, $S = (0,1)$), which cooperates when paired with a recipient with a good reputation and defects against a recipient with a bad reputation. We also relax this assumption and report qualitatively similar results for the continuous strategy space $(p,q) \in [0,1]^2$. (We neglect anti-discriminators $S = (1,0)$, which can never achieve high frequency.)

Players' reputations in the eyes of each member of the society are updated according to a social norm. In general, the update rule prescribed by a social norm can depend on the entire history of donor-recipient interactions, including the reputations of all interacting parties (*Santos et al., 2018*). Complex rules of moral evaluation, however, require high cognitive ability and effort that seem unrealistic in many real-world social interactions. Moreover, relatively simple 'second-order' norms of moral assessment, which update a donor's reputation based solely on the donor's action and the recipient's reputation, tend to outperform more complex social norms (*Santos et al., 2018*).

We consider second-order social norms, which can be encoded by a binary matrix $N_{ij}$. The row-index $i$ indicates donor's action, $i = 1$ for defect or $i = 2$ for cooperate; and the column-index $j$ indicates reputation of the recipient, $j = 1$ for bad or $j = 2$ for good. We focus on the four second-order norms that are most prominent in the literature: Stern Judging $\begin{pmatrix} G & B \\ B & G \end{pmatrix}$, Simple Standing $\begin{pmatrix} G & B \\ G & G \end{pmatrix}$, Scoring $\begin{pmatrix} B & B \\ G & G \end{pmatrix}$, and Shunning $\begin{pmatrix} B & B \\ B & G \end{pmatrix}$. For example, under Stern Judging (SJ) or Simple Standing (SS) an observer will assign a good reputation to a donor who punishes a recipient with a bad reputation, by defection. Whereas under Shunning (SH) or Scoring (SC) an observer will assign a bad reputation to a donor who defects against any recipient, regardless of the recipient's reputation. Following (*Sasaki et al., 2017*) we also allow for errors in strategy execution and in observation: a cooperative act is erroneously executed as defection with probability $e_1$, while an observer erroneously assigns a bad reputation instead of a good reputation, and *vice versa*, with probability $e_2$.

The broad consensus in the literature is that Stern Judging is the most efficient norm for promoting cooperation, along with widespread adoption of the discriminator strategy. This result is robust to variation in strategy exploration rates (*Santos et al., 2016a*), population sizes and error rates (*Santos et al., 2016b*), and it even extends to the realm of more complex norms of third and fourth order (*Santos et al., 2018*). *Pacheco et al. (2006)* have additionally shown that Stern Judging is the norm most likely to evolve in a group-structured population, because it maximizes the collective payoff of the group.

Prior studies of cooperation and moral assessment (*Nowak and Sigmund, 1998*; *Pacheco et al., 2006*; *Ohtsuki et al., 2009*; *Sasaki et al., 2017*; *Santos et al., 2016a*; *Santos et al., 2016b*) have assumed that reputations are objective and common knowledge in the population – meaning that opinions about reputations do not differ among individuals. Here we relax this assumption and allow individuals to differ in their opinions about one another. This reveals an under-appreciated subtlety in the application of norms for updating reputations. Namely, when an observer updates the reputation of a donor interacting with a recipient, the 'reputation of the recipient' could be considered either from the observer's own perspective, or from the donor's perspective. Under a purely egotistical application of a social norm, the 'recipient reputation' means the reputation in the eyes of the observer, who is forming an assessment of the donor. In this case the observer either ignores, or is unaware of, the donor's view of the recipient. This case corresponds to $E = 0$ in our analysis, the no-empathy model of moral assessment. However, we also analyze the possibility of empathetic moral assessment, $E > 0$, whereby the observer may account for donor's view of the recipient's reputation when assessing the donor. In the extreme case $E = 1$, for example, the observer always uses the donor's view of the recipient's reputation when applying the social norm to update the donor's reputation. In general, the parameter $E \in [0,1]$ determines the probability that an observer uses the donor's view of the recipient's reputation, as opposed to her own view, when applying the social norm to update the donor's reputation (see *Figure 1* and *Equations 5–8* in Materials and methods).

## Results

### Empathetic moral judgment facilitates cooperation

To analyze how empathy influences cooperation we first examine strategy evolution with a fixed degree of empathy $0 \leq E \leq 1$. We use the classic replicator-dynamic equations (*Taylor and Jonker, 1978*; *Nowak and Sigmund, 2004*) that describe how the frequencies of strategies (ALLD, ALLC, and DISC) evolve over time in an infinite population of players' strategies reproducing according to their payoffs. To simplify analysis, in the infinite-population model we assume that reputation frequencies reach equilibrium before strategies are updated – that is, the timescale of reputation updating is faster than that of strategy evolution (see Materials and methods). For each of the four most common norms we find bi-stable dynamics (*Sasaki et al., 2017*). That is, depending on the initial conditions the population will evolve to one of two stable equilibria: a monomorphic population of pure defectors, which supports no cooperation, or a population of cooperative (non-ALLD) strategies that supports some degree of cooperation.

How does empathy influence the prospects for cooperation? Under the Scoring norm, strategic evolution does not depend on the degree of empathy, because this norm ignores the recipient's reputation when updating a donor's reputation. For the other norms considered, however, empathy tends to increase cooperation. In particular, the basin of attraction towards the stable equilibrium that supports cooperation (green regions in *Figure 2*) is always larger when players are more empathetic – meaning that when $E$ is larger, there is a larger volume of initial conditions in the strategy space that lead to the stable equilibrium supporting cooperation.

In the case of Shunning and Stern judging, the stable equilibrium that supports cooperation consists of a monomorphic population of discriminators (*Figure 2*). Not only is the basin of attraction towards this equilibrium larger when a population is more empathetic, but so too is the equilibrium frequency of cooperative actions increased by greater degrees of empathy (*Figure 2* and *Figure 2—figure supplement 1*). And so empathy increases the frequency of outcomes that support cooperation, and also increases the frequency of cooperation at these outcomes.

In the case of Simple Standing the stable equilibrium that supports cooperation consists of a mix of ALLC and DISC strategists. The discriminator frequency at this equilibrium increases with empathy as

$$f_Z^\star = \frac{1}{s^2(1-e_2) - s + 1 - \varepsilon}\left(1 - \varepsilon + \frac{s(2-\varepsilon-e_2)-1}{(1-E)(\varepsilon-e_2)}\right) \tag{1}$$

where $\varepsilon = (1-e_1)(1-e_2) + e_1 e_2$ and $s = b/c$, until it reaches $f_Z^\star = 1$. The rate of cooperative play at this mixed equilibrium shows only a weak dependence on the degree of empathy (*Figure 2—figure supplement 1*).

Aside from the stable equilibria discussed above, for all four norms there is also an unstable equilibrium, with some portion of the population playing ALLD and some portion playing DISC. The frequency of discriminators at this unstable equilibrium is

$$f_Z = \frac{c}{b(\varepsilon - e_2)}(1 + (1-E)\gamma_{\mathrm{Norm}}), \tag{2}$$

where

$$\gamma_{\mathrm{SH}} = \frac{s^2(1-e_2) - s(1 + e_2(\varepsilon - e_2)) + e_2}{e_2(s^2-1) + E(s^2(1-e_2) - s + e_2)},$$

$$\gamma_{\mathrm{SJ}} = \frac{(s^2+1)(1-e_2) - s - s(1-e_2)(\varepsilon - e_2)}{E((s^2+1)(1-e_2) - s) - 1 + e_2 + s/2},$$

$$\gamma_{\mathrm{SS}} = \frac{(s^2+1)(1-e_2) - s - s(\varepsilon - e_2)(1-e_2)}{E((s^2+1)(1-e_2) - s) + (s^2-1)(1-e_2)}.$$

When $E = 1$ this expression coincides with the expressions found by *Sasaki et al. (2017)*. This result reflects the sense in which previous studies that assumed no variation in personal opinions about reputations are mathematically equivalent to always taking another person's perspective ($E = 1$).

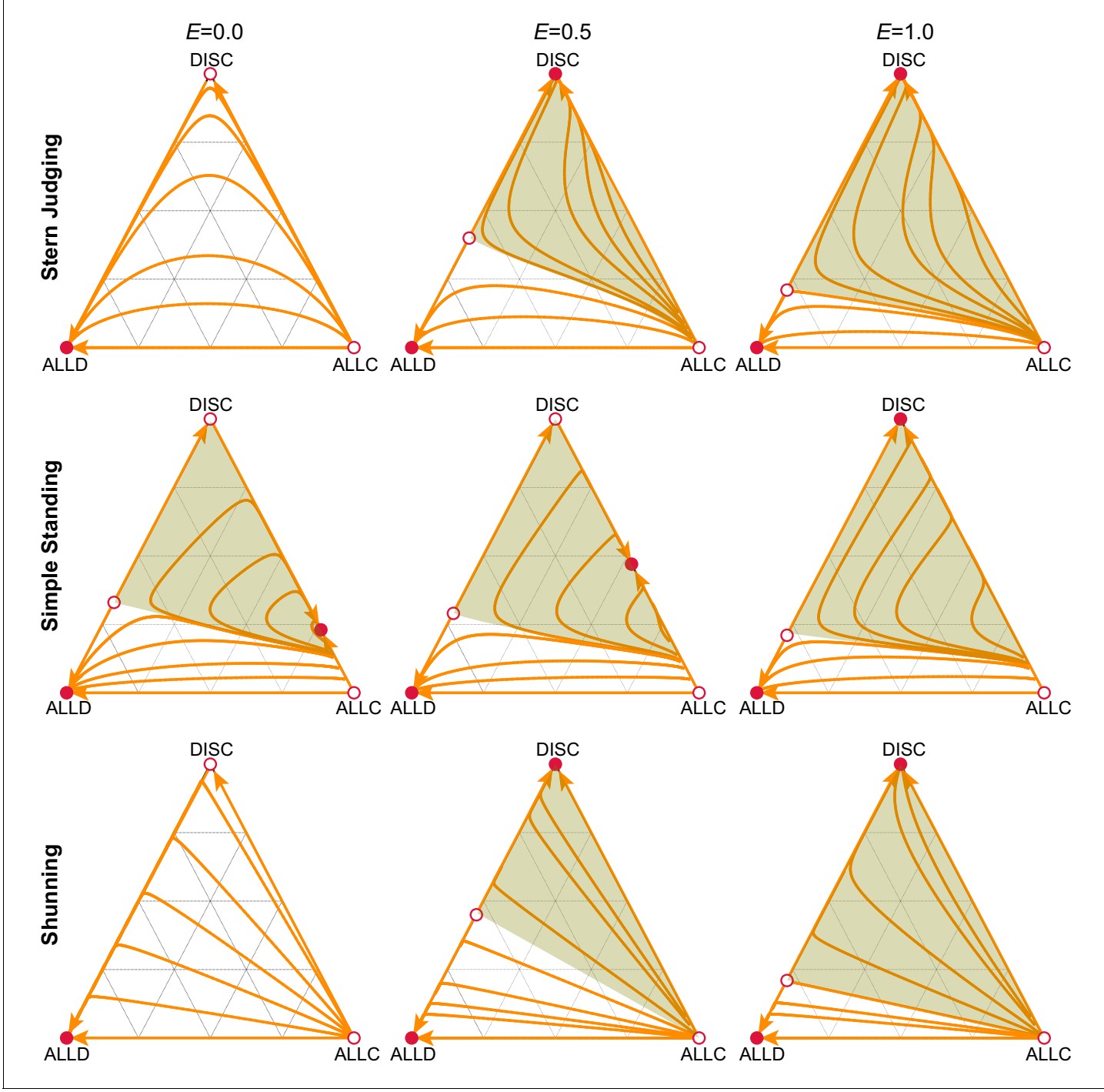

**Figure 2.** Empathetic moral evaluation facilitates the evolution of cooperation. We analyzed strategy evolution in the donation game under different social norms of moral assessment. Triangles describe the frequencies of three alternative strategies: unconditional defectors (ALLD), unconditional cooperators (ALLC), and discriminators (DISC) who cooperate with good recipients and defect against bad recipients. Red circles indicate the stable (filled) and unstable (open) strategic equilibria under replicator dynamics. The basin of attraction towards a stable equilibrium that supports cooperation (green) is larger as empathy, $E$, increases, for all three social norms shown. Orange curves illustrate sample trajectories towards stable equilibria. Costs and benefits are $c = 1.0$, $b = 5.0$, and error rates are $e_1 = e_2 = 0.02$.

DOI: https://doi.org/10.7554/eLife.44269.004

The following figure supplement is available for figure 2:

**Figure supplement 1.** Cooperation rates at the cooperative equilibria of strategy evolution.

DOI: https://doi.org/10.7554/eLife.44269.005

## Social norms that promote cooperation

In a finite population the frequencies of strategies do not evolve towards a fixed stable equilibrium, but rather continue to fluctuate, irrespective of initial conditions, due to demographic stochasticity. To study the impact of empathy on cooperation in this setting we undertook Monte Carlo simulations. In this model, successful strategies spread through social contagion: a strategy is copied with the probability $1/(1 + \exp(-w[\Pi_1 - \Pi_0]))$, where $w$ is the selection strength, and $\Pi_1$ and $\Pi_0$ are payoffs of two randomly selected individuals (*Traulsen et al., 2007*; *Traulsen et al., 2010*, see Materials and methods). In addition to these imitation dynamics, player strategies also change via random exploration at a rate $\mu$. For the sake of simplicity, we assumed that the timescale at which games are played and payoffs are acquired is much faster than the timescales of imitation, exploration, and reputation dynamics, so that each individual plays many games selection and mutation take place (see Materials and methods).

Empathy tends to increase mean levels of cooperation in finite populations under stochastic dynamics (*Figure 3*), similar to our findings in an infinite population. The effects of empathy are pronounced: the stationary mean frequency of cooperation ranges from near zero to near unity, in response to increasing the value of the empathy parameter $E$.

For high values of empathy, Stern Judging is the most efficient social norm at promoting cooperation, followed by Simple Standing and Scoring. This rank ordering of social norms is consistent with the prior literature (*Santos et al., 2016a*; *Santos et al., 2016b*; *Santos et al., 2018*). However we find a striking reversal from the established view of social norms when individuals are less empathetic. As $E \rightarrow 0$ Scoring promotes the most cooperation, while Stern Judging and Simple Standing engender less cooperation. And so the level of empathy strongly influences the amount of cooperation that evolves, and it even changes the ordering of which social norms are best at promoting cooperation. In particular, Stern judging is the most socially beneficial norm (*Pacheco et al., 2006*; *Santos et al., 2016a*; *Santos et al., 2016b*; *Santos et al., 2018*) only when individuals account for subjectivity in moral assessment, or when individuals are forced to agree with one another through a centralized institution of moral assessment.

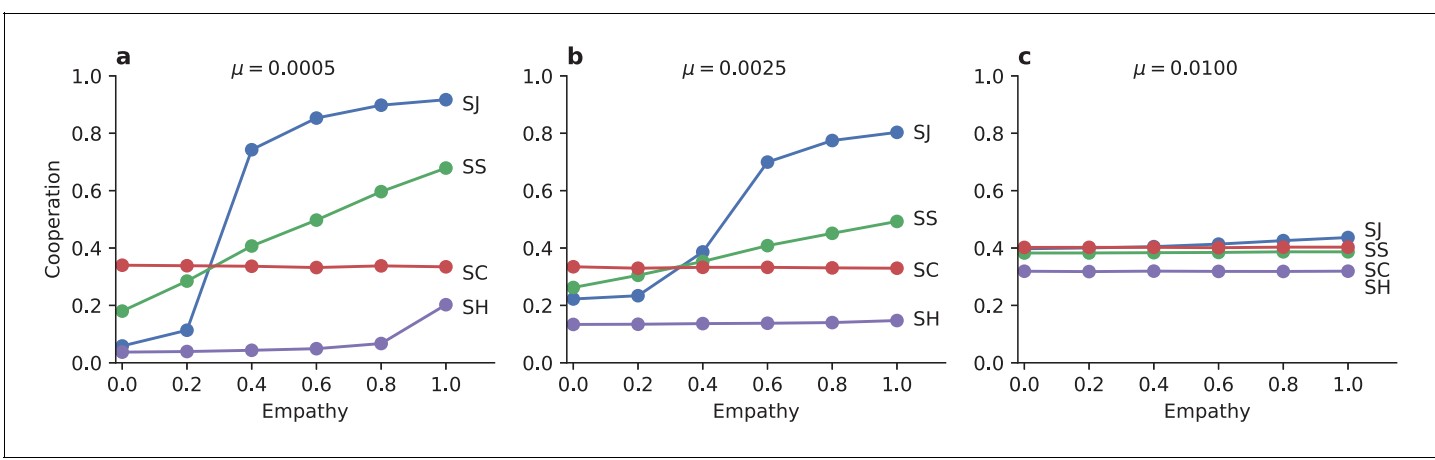

**Figure 3.** Empathetic moral judgment facilitates cooperation. The degree of empathy, $E$, determines which social norms of moral assessment produce the most cooperation and thus the greatest social benefit. (a) The Stern Judging (SJ) norm supports the highest rate of cooperation when empathy is high. But Stern Judging performs poorly under egocentric moral judgment, where Scoring (SC) and Simple Standing (SS) produce greater levels cooperation. The scoring norm (SC) does not depend on reputations, and so it shows cooperation levels that are insensitive to the level of empathy. The Shunning norm (SH) always produces the lowest level of cooperation. (b, c) As the strategy exploration rate $\mu$ increases, Stern Judging and Simple Standing become less efficient at promoting cooperation under highly empathetic moral evaluation, but they perform better under egocentric evaluation. All panels show ensemble mean cooperation levels in replicate Monte Carlo simulations of $N = 100$ individuals. Similar results hold in a continuous strategy space (see *Figure 3—figure supplement 1*).
DOI: https://doi.org/10.7554/eLife.44269.006

The following figure supplement is available for figure 3:

**Figure supplement 1.** Mean cooperation rates with continuous strategies.
DOI: https://doi.org/10.7554/eLife.44269.007

## Evolution of empathy

We have seen that empathy promotes cooperation in finite populations with reputation-conditional strategies. However, empathy is not inevitable and not universal in humans (*Cikara et al., 2014*). It remains unclear if empathy itself can evolve to high levels, and whether a population of empathetic individuals can resist invasion from egocentric moral evaluators. In the following analysis we assume that the degree of empathy in moral evaluation can be observed, inferred or learned, and can therefore evolve through social contagion (imitation dynamics) (*Cushman et al., 2017*), similar to how social norms are learned (*Buckholtz and Marois, 2012*). Alternatively, an individual's capacity for empathetic observations may have a genetic component evolving via Darwinian selection.

We analyze the evolution of empathy using the framework of adaptive dynamics (*Geritz et al., 1998*). Assuming rare mutations to the continuous empathy trait $E \in [0, 1]$, we calculate the invasion fitness of an invader $E_{\mathrm{I}}$ in an infinite resident population with empathy $E_{\mathrm{R}}$ by comparing their expected payoffs. We report pairwise-invisibility plots and investigate the evolutionary stability of singular points $E^*$, where the gradient of invasion fitness $\partial W(E_{\mathrm{R}}, E_{\mathrm{I}})/\partial E_{\mathrm{I}}$ (evaluated at $E_{\mathrm{I}} = E_{\mathrm{R}}$) vanishes. To support our analytic treatment we also perform Monte Carlo simulations in finite populations subject to demographic stochasticity, where empathy evolves through social copying according to individual payoffs, similar to strategy evolution under imitation dynamics (*Traulsen et al., 2007*).

Evolution can often favor empathy, depending upon the social norm and the initial conditions. To study empathy dynamics, we initially assume that the population is monomorphic for the discriminator strategy. In the case of the Shunning norm, then, there is a single, repulsive singular value of empathy (*Figure 4e*) at

$$E_{\mathrm{SH}}^* = \frac{e_2}{c/b + e_2 - 1}\left(1 - \frac{c/b}{\varepsilon - e_2}\right) + \frac{c/b}{\varepsilon - e_2}. \tag{3}$$

Such a population is bistable. If the initial level of empathy exceeds $E_{\mathrm{SH}}^*$ the population will evolve towards complete empathy ($E = 1$) and the discriminator strategy will remain stable; but if the initial level of empathy is less than $E_{\mathrm{SH}}^*$ the population will evolve towards complete egocentrism ($E = 0$), at which point the discriminator strategy is no longer stable (*Figure 2g*) and the population will be replaced by pure defectors. The singular value $E_{\mathrm{SH}}^*$ decreases as the benefit of cooperation $b/c$ increases, permitting a larger space of initial conditions that lead to the evolution of complete empathy (*Figure 5c*). And so, in summary, under the Shunning norm long-term strategy and empathy co-evolution will tend towards a completely empathetic population of discriminators, especially when the benefits to cooperation are high; or, alternatively, evolution will lead to complete population-wide defection.

Similar dynamics occur under the Stern Judging norm. In this case, starting from a monomorphic population of discriminators, there are two singular values for $E$: an evolutionary repeller $E_{\mathrm{SJ}}^* < 1/2$ and attractor $E_{\mathrm{SJ}}^* > 1/2$ (*Figure 4a,b*) given by

$$E_{\mathrm{SJ}}^* = \frac{1}{2} \pm \sqrt{\frac{1}{4} + \left(\frac{(1 - c/b)(e_2 + \varepsilon - 1 - c/b)}{(\varepsilon - e_2)^2}\right)}. \tag{4}$$

Provided empathy initially exceeds the repulsive value evolution will favor increasing empathy towards the attractive value, and the population of discriminators will remain stable. Increasing $b/c$ again favors the evolution of empathy, as it increases the value of the locally stable $E_{\mathrm{SJ}}^* > 1/2$ and also the range of initial values that that lead to $E_{\mathrm{SJ}}^* > 1/2$ through fixation of small mutations (*Figure 5a*). However, if empathy starts below the repulsive value, selection will favor evolution toward the attractive singular point at $E_{\mathrm{SJ}}^* = 0$, which no longer supports DISC as a stable equilibrium in strategy space (*Figure 2a*). And so, in summary, under Stern Judging co-evolution of strategies and empathy will tend towards a highly empathetic population of discriminators, especially when the benefits to cooperation are large; or, alternatively, evolution will lead to all defectors and empathy will thereafter drift neutrally.

The evolution of empathy is more complicated under Simple Standing. Assuming the population consists of discriminators there is a single evolutionarily stable and globally attractive singular point

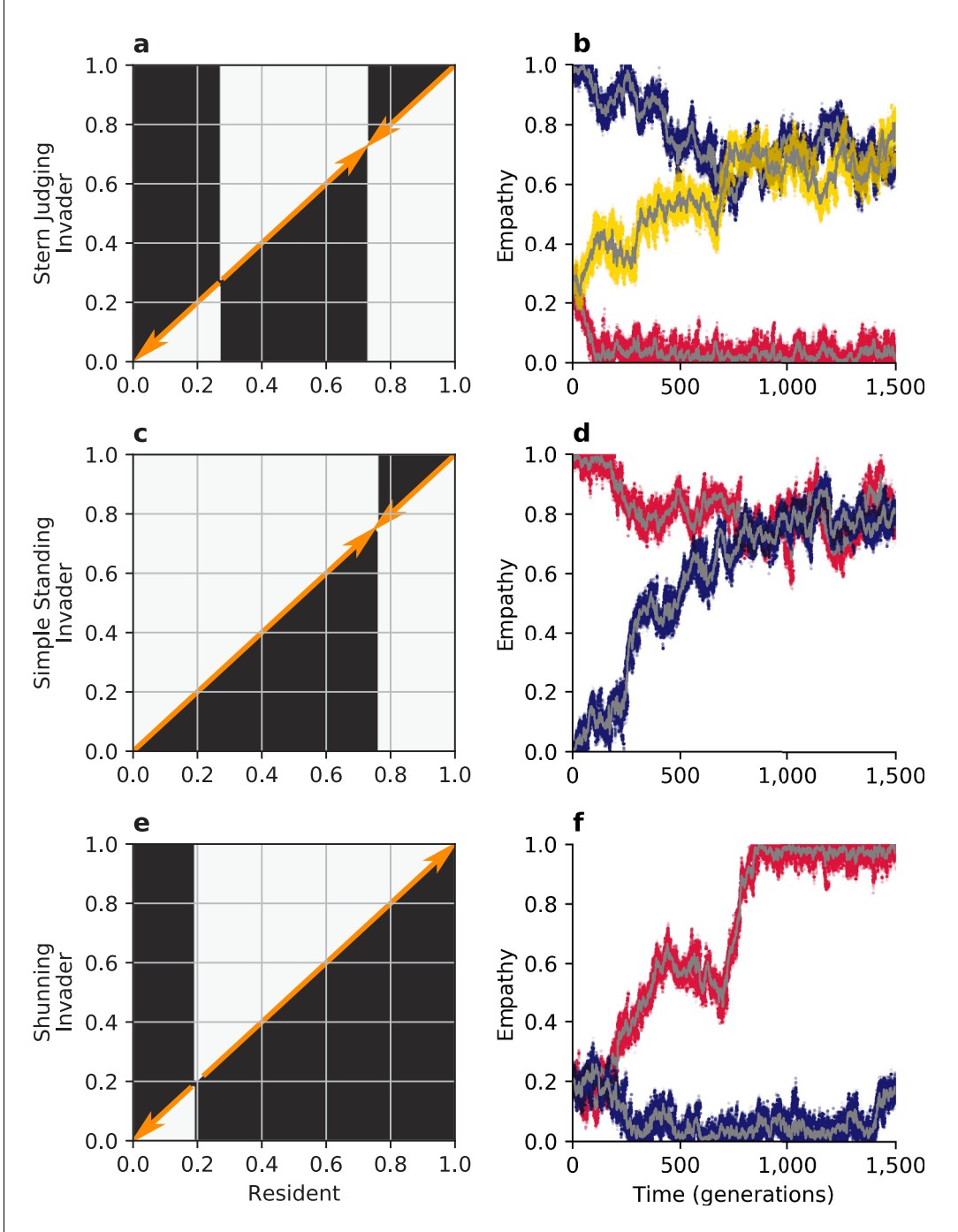

**Figure 4.** Evolution of empathy. The figure summarizes analytical predictions for empathy evolution under three different social norms, using adaptive dynamics in a population of discriminators, compared to Monte Carlo simulations in finite populations. (a, c, e) White areas in the pairwise invasibility plots show values of $E$ for which the invader's expected payoff exceeds the mean payoff of the resident population. Orange arrows indicate the direction of predicted evolution. (b, d, f) Monte Carlo simulations in small populations of 100 individuals with recurring mutations to $E$ reflect the predictions of adaptive-dynamics analysis. For each norm, sample trajectories showing all $E$ values in three independent populations are show in colors (red, blue, yellow), with the population means shown in gray.
DOI: https://doi.org/10.7554/eLife.44269.008

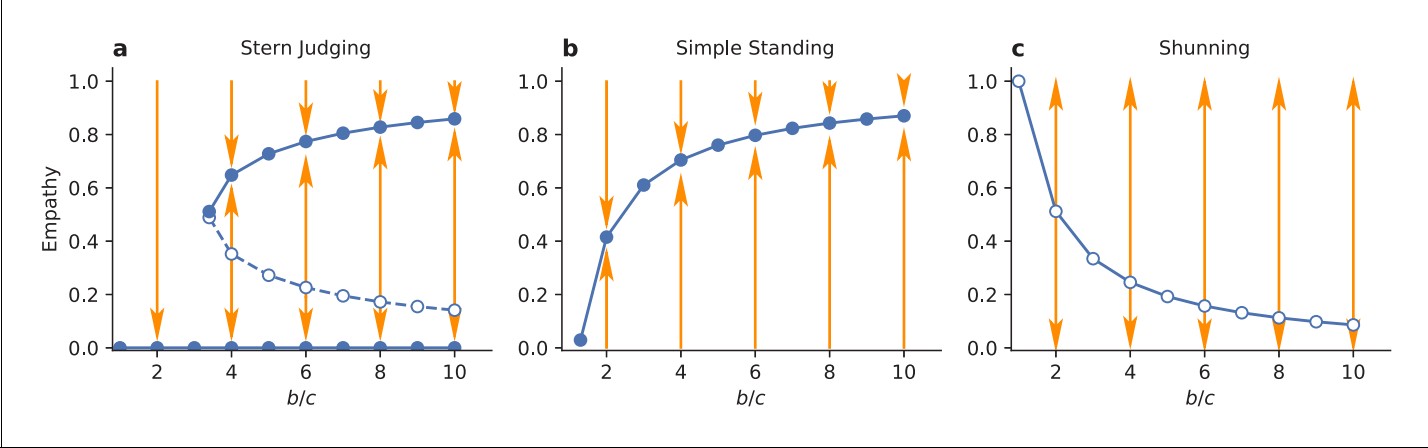

**Figure 5.** Evolutionary stability of empathy. Circles indicate evolutionarily stable (solid) and unstable (open) singular values of empathy, $E$, in an infinite population of discriminator strategists. (a) Above a critical benefit-cost ratio $b/c$, increasing the benefit of cooperation promotes the evolution of high levels of empathy under Stern Judging norm. (b) Under Simple Standing there is a single ESS value for empathy. The highest levels of empathy evolve with high benefits and low costs of cooperation. However, in this case the monomorphic discriminator equilibrium is not stable at the ESS value of empathy. (c) In populations governed by the Shunning norm there are no stable internal equilibria for $E$, and empathy will evolve to either one or zero.
DOI: https://doi.org/10.7554/eLife.44269.009

The following figure supplement is available for figure 5:

**Figure supplement 1.** Empathy-strategy co-evolution in an infinite population under Simple Standing norm.
DOI: https://doi.org/10.7554/eLife.44269.010

$E^*_{\text{SS}}$ (*Figure 4c*). The value of empathy at this singular point is larger when the benefits of cooperation are larger (*Figure 5b*). However once this value of empathy is reached, the strategic equilibrium at pure discriminators is no longer stable, and the population will instead be replaced by a mix of DISC and ALLC strategists, under replicator dynamics. This new strategic equilibrium will, in turn, lower the singular value of $E^*_{\text{SS}}$ under adaptive dynamics, which again changes the equilibrium balance of DISC and ALLC strategists. Long-term strategy-empathy co-evolution will continue in this fashion, with both ALLC and DISC present in the population, until the singular value of empathy reaches $E^*_{\text{SS}} = 0$ (see *Figure 5—figure supplement 1*). The strategic equilibrium at this point lies near the boundary of two basins of attraction (*Figure 2d*) and is vulnerable to invasion by pure defectors in a finite population. And so, in summary, while the exact dynamics will depend on the time scales of empathy and strategy evolution, Simple Standing cannot sustain empathy over the long term as both these components of personality co-evolve, eventually resulting in a population of pure defectors.

## Discussion

Empathy has long been associated with prosocial behavior and altruism in humans. Much of the existing literature focuses on the emotional component of empathy – linkage of emotional states between individuals, emotional contagion (*Hatfield et al., 1993*) and empathy-induced helping (*Cialdini et al., 1997*; *May, 2011*). For instance, there is substantial evidence that the effect of 'self-other merging' provides moral motivation to cooperate (*Batson et al., 1997*; *Batson et al., 1995*; *Batson and Moran, 1999*) and contributes to the resolution of public-goods dilemmas (*Batson, 1994*). Empathy is not a unitary construct, however, and besides the purely emotional reaction to the states of other individuals there is the cognitive ability to understand another person's psychological perspective (*Davis, 1983*; *Smith, 2006*).

Very little is known about the origins of empathy in relation to cooperative behavior, although some research suggests that the capacity for emotional empathy evolved in the context of parental care (*de Waal, 2008*). Even less is known about social evolution and selective forces operating on empathy in modern societies. Even if empathy promotes altruistic behavior, why should empathetic perspective-taking itself evolve and be stable against the invasion of morally egocentric individuals?

Here we have studied the role of empathetic perspective-taking in a game-theoretic context of moral evaluation, where individuals make moral judgments from their own subjective perspectives. Studying the impact of empathy in this context is critical to understanding cooperative behavior in modern, highly-connected societies that generally lack a centralized institution of objective moral assessment.

Social norms specify the rules of moral evaluation. It is well known that moral reputations can sustain high levels of cooperation if individuals discriminate between the 'good' and the 'bad'. Social norms themselves likely emerge from individual beliefs of what reputations should be assigned to defectors and cooperators in distinct social situations. While some studies assume that social reputations are absolute – for instance due to shared information, public monitoring, institutions and gossip – our study draws attention to the potential for disagreements on reputations that arise from errors or different observation histories. The same individual can have different reputations in the eyes of distinct observers; in other words, moral evaluations are not absolute, and social reputation is relative. When monitoring of social interactions is private, cooperation is much harder to evolve and sustain – as reflected by the results of two recent studies by *Hilbe et al. (2018)* and *Okada et al. (2017)*. Both of those studies analyzed models with private, but egocentric, moral evaluation corresponding to $E = 0$ in our analysis. Our model of private evaluation makes the additional assumption that each observer evaluates a donor based on a different social interaction, along the lines of *Okada et al. (2018)*.

Our key finding is that high levels of cooperation can be sustained, even with private monitoring of reputations, provided individuals recognize moral relativity and are capable of making moral judgments from another person's perspective ($E > 0$). Egocentric evaluation leads to unjustified or irrational defection, because a person perceived as 'bad' by the observer might actually appear 'good' in the eyes of the donor who's action is being evaluated, or *vice versa*. This point is particularly striking in the case of Stern Judging, the norm that assigns a 'good' reputation only to individuals who cooperate with other 'good' players and defect against 'bad' (*Kandori, 1992*; *Pacheco et al., 2006*). Despite being the most efficient norm at promoting cooperation in empathetic societies, Stern Judging performs very poorly in egocentric populations. On the other hand, Scoring – the norm that does not take into account the recipient's reputation at all – is immune to the effects of empathy and dominates in societies with egocentric moral evaluation rules.

Finally, we have shown that empathetic perspective-taking can evolve through cultural copying, and remain evolutionarily stable if a society is governed by Stern Judging or Shunning. Once these societies evolve empathy, individuals performing egocentric evaluations of observed social behavior will be rewarded less than their empathetic peers, and this remains true even if strategies are allowed to co-evolve with empathy. However, we have also seen that egocentric and uncooperative societies are nevertheless possible evolutionary outcomes. In populations governed by Stern Judging, Shunning and Scoring this outcome represents an alternative locally (though not globally) attractive stable state in the strategy-empathy phase space. In the case of Simple Standing, the egocentric and uncooperative outcome is the only long-term stable outcome as both empathy and strategies are allowed to evolve.

Our study raises a number of questions to be addressed in future work on empathy, norms, and the evolution of cooperation. Whereas we have studied empathy as a fixed trait, an individual's tendency for empathetic evaluation might instead depend in a non-linear way on the current make-up of strategies in the population. Another question involves the competition of social norms for moral evaluation – a topic that has been studied in some contexts, such as when errors do not occur (*Uchida et al., 2018*), or in the presence of population structure (*Masuda, 2012*; *Pacheco et al., 2006*). Perhaps an even more fundamental question is whether and how population-wide social norms can evolve from individual moral beliefs to begin with. It is unclear whether social contagion or individual-level Darwinian selection is sufficient to establish a hierarchy of norms governing how individuals update each others' reputations in a population. We have shown that the norms that promote the most cooperation change depending on the capacity for empathetic perspective-taking, but should we also expect different norms to evolve under empathetic and egocentric modes of judgment? For instance, populations characterized by fully empathetic moral judgment might be conducive to the evolution of selfish norms that indiscriminately assign 'bad' reputations to evade costly cooperation without being punished, while models with private egocentric evaluation may lead to the evolution of more cooperative norms, such as Scoring or Stern Judging

(*Yamamoto et al., 2017*; *Uchida et al., 2018*). Such questions about the origin of and competition between moral norms remain outstanding.

## Materials and methods

### Cooperation under empathetic moral evaluation

#### Replicator dynamics

To analyze the evolutionary dynamics of strategies, we consider replicator dynamics in an infinite population with a fixed social norm and fixed value of the empathy parameter $E$, limiting ourselves to the three strategies: ALLC or $X$, ALLD or $Y$, and DISC or $Z$. Denoting the mean payoff of strategy $s$ as $\Pi_s$, and the frequencies of the three strategies at $f_s$, the strategy dynamics can be described as $\frac{df_s}{dt} = f_s(\Pi_s - \sum_s f_s \Pi_s)$. As in *Okada et al. (2018)*, we assume that each observer makes her moral evaluation of a donor's action based on a different social interaction, that is, each interaction is observed only once.

To describe image dynamics, we let $g$ denote the frequency of 'good' individuals within the population, that is $g = f_X g_X + f_Y g_Y + f_Z g_Z$. Furthermore we let $g_2$ denote the probability that two randomly selected individuals will see the same recipient as 'good', that is $g_2 = f_X g_X^2 + f_Y g_Y^2 + f_Z g_Z^2$. The probability that two individuals will see the same recipient as 'bad' is then $b_2 = 1 - 2g + g_2$, and they will disagree on the subjective reputation of the recipient with the probability $d_2 = g - g_2$.

For the Stern Judging norm, mean frequencies of 'good' individuals within the subpopulations of cooperators, defectors and discriminators are (averaged over the perspectives of all players):

$$
\begin{aligned}
g_X &= g\varepsilon + (1-g)(1-\varepsilon); \\
g_Y &= g e_2 + (1-g)(1-e_2); \\
g_Z &= E(g\varepsilon + (1-g)(1-e_2)) + (1-E)(g_2\varepsilon + d_2(e_2+1-\varepsilon) + b_2(1-e_2)).
\end{aligned}
\tag{5}
$$

Here $\varepsilon = (1-e_1)(1-e_2) + e_1 e_2$ is the probability that an individual with the intention to cooperate is assigned a 'good' reputation. The second term of $g_Z$ describes egocentric evaluation, in which the donor and the observer may disagree on the reputation of the recipient. If both see the recipient as 'good', the donor will cooperate under the DISC strategy, and she will be seen as 'good' in the eyes of the observer with the probability $\varepsilon$. If the donor sees the recipient as 'good', but the observer disagrees, the donor will intend to cooperate and will be seen as 'good' only with probability $1-\varepsilon$, that is, if cooperation succeeds but an evaluation error occurs $((1-e_1)e_2)$ or if cooperation fails and the action is evaluated correctly $(e_1(1-e_2))$). In the opposite case, the donor will defect and will be seen as 'good' only if the evaluation error $e_2$ happens. Finally, if the donor and the observer both see the recipient as 'bad', the donor will defect and will be seen as good only in the absence of observation error $1-e_2$.

Expected payoffs of the three strategies are then:

$$
\begin{aligned}
\Pi_X &= b(f_X + f_Z g_X)(1-e_1) - c(1-e_1); \\
\Pi_Y &= b(f_X + f_Z g_Y)(1-e_1); \\
\Pi_Z &= b(f_X + f_Z g_Z)(1-e_1) - cg(1-e_1); \\
\Pi &= f_X \Pi_X + f_Y \Pi_Y + f_Z \Pi_Z.
\end{aligned}
\tag{6}
$$

Likewise for Shunning:

$$
\begin{aligned}
g_X &= g\varepsilon + (1-g)e_2; \\
g_Y &= e_2; \\
g_Z &= E(g\varepsilon + (1-g)e_2) + (1-E)(g_2\varepsilon + 2d_2 e_2 + b_2 e_2).
\end{aligned}
\tag{7}
$$

For Simple Standing:

$$
\begin{aligned}
g_X &= g\varepsilon + (1-g)(1-e_2); \\
g_Y &= g e_2 + (1-g)(1-e_2); \\
g_Z &= E(g\varepsilon + (1-g)(1-e_2)) + (1-E)(g_2\varepsilon + d_2 + b_2(1-e_2)).
\end{aligned}
\tag{8}
$$

And finally for Scoring norm, empathy $E$ is irrelevant, because the norm does not take into account the reputation of the recipient:

$$\begin{aligned}
g_X &= \varepsilon; \\
g_Y &= e_2; \\
g_Z &= g\varepsilon + (1-g)e_2.
\end{aligned} \qquad (9)$$

## Stochastic simulations

In addition to the deterministic replicator-dynamics analysis of strategy evolution, we performed a series of individual-based simulations to measure mean levels of cooperation under continuous influx of mutations in the strategy space (*Santos et al., 2016a*). We assume that all individuals follow the same social norm and are characterized by the same value of empathy, $E$. The population consists of $N$ individuals, each with its own strategy and its own subjective list of reputations. Each generation, any given individual interacts with all other members of the society in three different roles: once as a donor, once as a recipient, and once as an observer.

First, each individual plays a single round of the donation game with all other members of the society according to her strategy $S = (p, q)$ and the subjective reputation of the recipient, also taking into account the implementation error $e_1$. Here $p$ and $q$ denote the probabilities that a donor cooperates with a 'bad' (B) and 'good' (G) recipient, respectively. The act of cooperation fails with the probability $e_1$ (defection always succeeds). The cumulative payoff is then assigned to each individual, with the benefit of cooperation fixed at $b$ and the cost of a cooperative act $c$.

To update their list of subjective reputations based on the social norm $N_{ij}$, each player then chooses to observe a single interaction per donor (that is, with a randomly chosen recipient), again taking into account subjective reputation of the recipient either in the eyes of the donor (probability $E$) or the eyes of the observer (with a probability $1 - E$). The newly assigned reputation is reversed with the probability $e_2$, representing observation errors. For the sake of simplicity, we assume that all reputations are updated simultaneously after all donor-recipient interactions have taken place.

We model selection and drift of strategies as a process of social contagion implemented as a pairwise comparison process. Following the reputation-updating step, a random pair of individuals is chosen; the first individual adopts the strategy of the second with the probability $1/(1 + \exp(-w[\Pi_1 - \Pi_0]))$, where $w$ is the selection strength, and $\Pi_1$ and $\Pi_0$ are payoffs of the two earned within the last generation. In our simulations of populations with $N = 100$ individuals, we used $w = 1.0$. Finally, each individual is subject to random strategy exploration, in which a new random strategy is adopted with a small probability $\mu$ (*Santos et al., 2016a*).

The simulation is initialized with random strategies and random lists of subjective reputations. We recorded the mean rate of cooperation averaged over 150,000 generations in 50 replicate populations, which is reported in *Figure 3*.

## Evolution of empathy

Let $g_{ij}$ be the frequency of 'good' individuals in the sub-population $i$ as seen by individuals belonging to the sub-population $j$, where $i$ and $j$ correspond either to resident ($i, j = 0$) or invader ($i, j = 1$) sub-population. Working in the limit of negligible invader frequencies, and assuming that the population consists only of DISC strategists, for Stern Judging norm we have:

$$\begin{aligned}
g_{00} &= E_0(g_{00}\varepsilon + (1 - g_{00})(1 - e_2)) \\
&\quad + (1 - E_0)\left(g_{00}^2\varepsilon + g_{00}(1 - g_{00})(e_2 + 1 - \varepsilon) + (1 - g_{00})^2(1 - e_2)\right); \\
g_{01} &= E_1(g_{00}\varepsilon + (1 - g_{00})(1 - e_2)) \\
&\quad + (1 - E_1)(g_{01}g_{00}\varepsilon + g_{01}(1 - g_{00})e_2 + (1 - g_{01})g_{00}(1 - \varepsilon) + (1 - g_{01})(1 - g_{00})(1 - e_2)); \\
g_{10} &= E_0(g_{01}\varepsilon + (1 - g_{01})(1 - e_2)) \\
&\quad + (1 - E_0)(g_{01}g_{00}\varepsilon + g_{01}(1 - g_{00})e_2 + (1 - g_{01})g_{00}(1 - \varepsilon) + (1 - g_{01})(1 - g_{00})(1 - e_2)).
\end{aligned} \qquad (10)$$

Here $\varepsilon = (1 - e_1)(1 - e_2) + e_1 e_2$, and $E_0$ and $E_1$ are empathy values of resident and invader sub-population. For Simple Standing norm, the relative frequencies of good individuals are:

$$
\begin{aligned}
g_{00} &= E_0(g_{00}\varepsilon + (1-g_{00})(1-e_2)) \\
&\quad + (1-E_0)\Big(g_{00}^2\varepsilon + g_{00}(1-g_{00})(1-e_2) + (1-g_{00})g_{00}e_2 + (1-g_{00})^2(1-e_2)\Big); \\
g_{01} &= E_1(g_{00}\varepsilon + (1-g_{00})(1-e_2)) \\
&\quad + (1-E_1)(g_{01}g_{00}\varepsilon + g_{01}(1-g_{00})e_2 + (1-g_{01})g_{00}(1-e_2) + (1-g_{01})(1-g_{00})(1-e_2)); \\
g_{10} &= E_0(g_{01}\varepsilon + (1-g_{01})(1-e_2)) \\
&\quad + (1-E_0)(g_{01}g_{00}\varepsilon + g_{01}(1-g_{00})(1-e_2) + (1-g_{01})g_{00}e_2 + (1-g_{01})(1-g_{00})(1-e_2)).
\end{aligned}
\tag{11}
$$

Likewise, for the Shunning norm:

$$
\begin{aligned}
g_{00} &= E_0(g_{00}\varepsilon + (1-g_{00})(e_2)) \\
&\quad + (1-E_0)\Big(g_{00}^2\varepsilon + g_{00}(1-g_{00})e_2 + (1-g_{00})g_{00}e_2 + (1-g_{00})^2 e_2\Big); \\
g_{01} &= E_1(g_{00}\varepsilon + (1-g_{00})e_2) \\
&\quad + (1-E_1)(g_{01}g_{00}\varepsilon + g_{01}(1-g_{00})e_2 + (1-g_{01})g_{00}e_2 + (1-g_{01})(1-g_{00})e_2); \\
g_{10} &= E_0(g_{01}\varepsilon + (1-g_{01})e_2) \\
&\quad + (1-E_0)(g_{01}g_{00}\varepsilon + g_{01}(1-g_{00})e_2 + (1-g_{01})g_{00}e_2 + (1-g_{01})(1-g_{00})e_2).
\end{aligned}
\tag{12}
$$

Under Scoring, the frequencies of 'good' individuals do not depend on empathy:

$$
g = \frac{e_2}{1-\varepsilon+e_2}.
\tag{13}
$$

We then calculate the expected payoffs of individuals in resident and invader sub-populations:

$$
\begin{cases}
\Pi_0 = b(1-e_1)g_{00} - c(1-e_1)g_{00}; \\
\Pi_1 = b(1-e_1)g_{10} - c(1-e_1)g_{01}.
\end{cases}
\tag{14}
$$

These payoffs are used to generate pairwise invasibility plots in *Figure 4*. Singular points are found by setting $\partial(\Pi_1 - \Pi_0)\partial E_1 = 0$ and setting $E_0 = E_1$.

## Individual-based simulations of empathy evolution

To verify the ESS results of the adaptive-dynamics calculations we performed a series of Monte-Carlo simulations in finite populations of $N = 100$ individuals. The simulation routine is largely the same as for strategy evolution, except that in this case we fixed the strategy at DISC and allowed $E$ to evolve via constant influx of small mutations. Each generation, empathy of an individual changes via mutation at a rate $\mu_E = 0.005$. Since empathy is a continuous parameter, we draw the mutational deviation $\delta E$ from a normal distribution centered around $\delta E_0 = 0$ with a standard deviation $\sigma = 0.01$. Selection for empathy is modeled by choosing five random pairs of individuals and assuming that in each pair the first individual copies the empathy value $E_1$ of the second with the probability $1/(1 + \exp(-w[\Pi_1 - \Pi_0]))$, where $\Pi_1$ and $\Pi_0$ are their payoffs.

## Additional information

### Funding

| Funder | Grant reference number | Author |
| --- | --- | --- |
| David and Lucile Packard Foundation | | Joshua B Plotkin |
| Army Research Office | W911NF-12-R-0012-04 | Joshua B Plotkin |

The funders had no role in study design, data collection and interpretation, or the decision to submit the work for publication.

## Author contributions
Arunas L Radzvilavicius, Conceptualization, Formal analysis, Writing—original draft; Alexander J Stewart, Conceptualization, Writing—review and editing; Joshua B Plotkin, Conceptualization, Methodology, Writing—review and editing

## Author ORCIDs
Joshua B Plotkin  http://orcid.org/0000-0003-2349-6304

## Decision letter and Author response
Decision letter https://doi.org/10.7554/eLife.44269.014
Author response https://doi.org/10.7554/eLife.44269.015

## Additional files

### Supplementary files
• Transparent reporting form
DOI: https://doi.org/10.7554/eLife.44269.011
• Source data 1. Zip folder with data for all figures, code to produce the figures from these data, and the simulation code that generated the data.
DOI: https://doi.org/10.7554/eLife.44269.012

### Data availability
The data for all figures, code to produce the figures from these data, and the simulation code that generated the data are provided as Source data 1.

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
