## [Decision Letter]

Thank you for submitting your article "Evolution of empathetic moral evaluation" for consideration by *eLife*. Your article has been reviewed by three peer reviewers, one of whom is a member of our Board of Reviewing Editors, and the evaluation has been overseen by Diethard Tautz as the Senior Editor. The following individuals involved in review of your submission have agreed to reveal their identity: Naoki Masuda (Reviewer #2); Martin A Nowak (Reviewer #3).

The reviewers have discussed the reviews with one another and the Reviewing Editor has drafted this decision to help you prepare a revised submission.

The paper analyses a mathematical model of reputation-based indirect reciprocity introducing the concept of empathy, which involves assessing the reputation of a recipient from the donor's perspective rather than from the assessors perspective. This is a very interesting and novel addition to the literature on evolutionary game theory in the context of cooperation and social norms. Before the paper can be published, however, we ask you to address a number of issues that the reviewers have pointed out.

*Reviewer #1:*

A few remarks regarding the adaptive dynamics of empathy:

I think the authors should point out that there is a certain degree of degeneracy in their model, which is due to the assumption that payoffs are linear in the empathy E (a common assumption in game theory, of course). This degeneracy is for example reflected in the vertical lines in the pairwise invasibility plots in Figure 4. It implies that there can for example be no evolutionary branching points in their model.

Generally, speaking, it is easy to imagine that empathy enters in a non-linear way into the model (e.g. when the empathy that an individual displays depends on population level quantities, such as the overall level of cooperation, or the average empathy, or when the empathy of individual is itself a (non-linear) function of some other underlying personal characteristic, etc.). In such cases, it is possible for diversity in empathy to evolve, which would be an interesting and perhaps realistic feature of the model.

Note for example that even in the present model, different levels of empathy can coexist "ecologically". For example, it is evident from Figure 4A that empathy levels 0.6 and 0.9 can invade each other, and hence would coexist. (However, if such coexisting strains were allowed to evolve, the would both evolve to the ESS value, and hence diversity would be lost over evolutionary time.)

Overall, I think it would be good if the paper would discuss these issues. Also, please note that in the second paragraph of the section "Evolution of empathy", the gradient derivative of the function W has to be evaluated at *E*_I_ = *E*_R_ to become the selection gradient.

*Reviewer #2:*

The authors analysed a mathematical model of reputation-based indirect reciprocity analytically (using replicator dynamics and adaptive dynamics) and numerically. Each player is assumed to be either unconditional cooperator, discriminator, or unconditional defector, and also assigns a binary reputation to other players which is updated in the course of game dynamics. This is a common setting (including the analysis methods) since early 2000s. A new component in this paper is the effect of empathy. The authors' model assumes that different players may have different opinions towards a player. This has been done in some prior modelling, but remarkably, the present authors assumed that an observer uses the donor's view (as opposed to the observer's own view) towards the recipient's reputation with probability E, when judging if the donor's action towards the recipient is good or bad. In this way the authors implemented the effect of empathy. This is realistic and presents an implementation of the theory of mind, thus stretching the implication of the present work to much wider domains such as social psychology and neuroscience, which many of similar studies lack. The authors discovered that empathy generally promotes cooperation. The effect of the empathy depended on the social norm (i.e. reputation assignment rule) employed by the population. Furthermore, the authors studied evolution of the empathy (i.e. value of E) both analytically and numerically.

The work presents a solid framework for analysing empathy, which is really a salient issue, in the setting of indirect reciprocity and has led to reasonable results. Furthermore, I feel that the work is very complete. Therefore, I recommend publication of this work.

*Reviewer #3:*

The paper explores the evolution of cooperation in a model of indirect reciprocity with empathy. It assumes that players are repeatedly matched to interact in a social dilemma game. A player's decision whether to cooperate may depend on the recipient's reputation. There are three possible strategies: ALLC players always cooperate, irrespective of the reputation of the recipient. ALLD players always defect. Discriminators cooperate if the co-player has a good reputation and they defect if the co-player has a bad reputation.

To assign reputations to players, the paper assumes that the entire population applies the same social norm. Four different social norms are considered, Stern Judging, Simple Standing, Scoring, and Shunning. All of these norms assign a good reputation to a donor who cooperates with a good recipient, and they assign a bad reputation to a donor who defects against a good recipient. The norms differ in how they assess interactions with a bad recipient.

As their main contribution, the authors study the effect of empathy. In traditional models of indirect reciprocity, observers judge the donor's action based on their own perspective of the recipient. In contrast, an empathetic observer judges the donor's action based on the donor's perspective of the recipient. Using a mixture of analytical methods and simulations, the main results are as follows:

1) Empathetic judgment facilitates the evolution of cooperation

2) Empathy itself can evolve for three of the four social norms studied.

I think the questions explored in this paper are extremely interesting for at least two reasons. First, a few recent studies have found that indirect reciprocity has problems to evolve when information is private. This paper proposes an elegant solution to this problem.

Second, empathy itself is an important characteristic of human behavior that requires an evolutionary explanation. The present paper can serve as an important first step.

The paper is well motivated, the model is sound, and the figures provide a great illustration of the key results.

However, a few questions remain because the Materials and methods section is somewhat unclear. In particular, I was unable to verify whether the payoff formulas are correct, and whether the qualitative findings are robust with respect to minor changes in the model assumptions. If these remaining questions can be resolved, I strongly support publication.

1) It is unclear how the payoffs for the replicator dynamics model have been derived. In fact, the given formulas might be wrong.

So far it has been difficult to analyze the effects of private information analytically. Most previous models of indirect reciprocity have relied on simulations (such as the two cited papers by Okada et al. and Hilbe et al., [Okada, Sasaki and Nakai, 2017; Hilbe et al., 2018]). Analytical results have only been feasible in rather specific special cases, for example, Okada et al., 2018 "A solution for private assessment in indirect reciprocity using solitary observation".

The difficulties arise because models of private information need to calculate the probability that two randomly chosen discriminators both assign a good reputation to the recipient. This probability is difficult to compute because the opinions of two discriminators will neither be perfectly correlated nor are they completely independent.

Given these previous difficulties, I was surprised to see that the present paper can give simple expressions for the players' payoffs (Equations 5-9 in the Materials and methods section), which also hold for E=0. Unfortunately Equations 5-9 are not derived in sufficient detail, which makes it impossible to check whether they are correct.

In these formulas, g is the frequency of good individuals within the population, whereas the authors seem to use g^2^ for the probability that two individuals both assign a good reputation to a random recipient. This seems to presume complete independence of individual opinions.

I would like to ask the authors to provide a more detailed derivation of the payoff Equations 5-9. In particular, the authors should make clear how they circumvented the problems described in the above mentioned paper of Okada et al., 2018.

The same comment also applies to the formulas in the Materials and methods section on the evolution of empathy.

2) The paper is based on the assumption that all players in the population employ the same social norm. As the authors note in the last paragraph of their discussion, it is not clear whether cooperation remains stable if mutants were allowed to switch to a self-serving social norm (by assigning a bad reputation to everyone, and then defect against everyone). It appears that such a mutant is a major threat to all social norms studied herein, especially when the empathy probability is high.

I completely agree with the authors that a full analysis of such mutations is beyond the scope of the present paper. However, I would appreciate if the authors could give some basic argument why their results can be expected to be robust.

As one possibility, the authors could repeat the analysis in Figure 2 or Figure 3. But instead of assuming that ALLD and ALLC players apply the same social norm as the discriminators, they could assume that ALLD players always assign a bad reputation to everyone, whereas ALLC players always assign a good reputation to everyone. This would correspond to a case in which the social norms applied by the unconditional types are consistent with their actions.

If such an analysis turns out to be infeasible, I suggest the authors explain under which circumstances we can reasonably assume all members of a population to use the same social norm.

3) The results in Figure 3C are either wrong, or they require more explanation.

In this figure, the authors show that for a rather small mutation rate (*μ*=0.01), almost all social norms lead to similar cooperation rates and that cooperation rates are largely independent of the empathy parameter. There are two reasons why Figure 3C is somewhat odd:

i) It suggests that empathy can only help for extremely small mutation rates.

ii) It suggests that in the case of no empathy E=0, Stern Judging can yield roughly 40% cooperation. This is somewhat surprising, given that previous papers have found that under private information, a Stern Judging norm typically leads to an ALLD population with almost no cooperation at all (e.g. Okada, Sasaki and Nakai, 2017; Hilbe et al., 2018).

On a similar note, I would appreciate if the authors could be more clear in the main text about the relevant timescales. From reading the Materials and methods section, it seems to me that the authors assume that in the infinite population model, reputations are assumed to reach a steady state before evolutionary updating occurs. In the finite population simulations, evolutionary updating may occur before the players' reputations have reached a steady state, is this correct?

---

## [Author Response]

Reviewer #1:A few remarks regarding the adaptive dynamics of empathy:I think the authors should point out that there is a certain degree of degeneracy in their model, which is due to the assumption that payoffs are linear in the empathy E (a common assumption in game theory, of course). This degeneracy is for example reflected in the vertical lines in the pairwise invasibility plots in Figure 4. It implies that there can for example be no evolutionary branching points in their model.Generally, speaking, it is easy to imagine that empathy enters in a non-linear way into the model (e.g. when the empathy that an individual displays depends on population level quantities, such as the overall level of cooperation, or the average empathy, or when the empathy of individual is itself a (non-linear) function of some other underlying personal characteristic, etc.). In such cases, it is possible for diversity in empathy to evolve, which would be an interesting and perhaps realistic feature of the model.Note for example that even in the present model, different levels of empathy can coexist "ecologically". For example, it is evident from Figure 4A that empathy levels 0.6 and 0.9 can invade each other, and hence would coexist. (However, if such coexisting strains were allowed to evolve, the would both evolve to the ESS value, and hence diversity would be lost over evolutionary time.)

Thank you for pointing this out. In this initial study, we model empathy in its simplest form as the probability to perform moral evaluations from the perspective of another person, which itself is a fixed trait of the focal individual. We agree that, in general, the propensity to make empathetic evaluations could depend on the state of the population in a non-linear way, leading to qualitatively different results including evolutionary branching. Although we leave this analysis for the future research, we have added some discussion about this in the final paragraph of the manuscript.

Overall, I think it would be good if the paper would discuss these issues. Also, please note that in the second paragraph of the section "Evolution of empathy", the gradient derivative of the function W has to be evaluated at E_I_ = E_R_ to become the selection gradient.

Thank you, we have appended this to the expression of selection gradient. In addition, we discuss the possibility for non-linearity in empathy in our Discussion section.

Reviewer #3:[…] However, a few questions remain because the Materials and methods section is somewhat unclear. In particular, I was unable to verify whether the payoff formulas are correct, and whether the qualitative findings are robust with respect to minor changes in the model assumptions. If these remaining questions can be resolved, I strongly support publication.1) It is unclear how the payoffs for the replicator dynamics model have been derived. In fact, the given formulas might be wrong.So far it has been difficult to analyze the effects of private information analytically. Most previous models of indirect reciprocity have relied on simulations (such as the two cited papers by Okada et al. and Hilbe et al., [Okada, Sasaki and Nakai, 2017; Hilbe et al., 2018]). Analytical results have only been feasible in rather specific special cases, for example, Okada et al., 2018 "A solution for private assessment in indirect reciprocity using solitary observation".The difficulties arise because models of private information need to calculate the probability that two randomly chosen discriminators both assign a good reputation to the recipient. This probability is difficult to compute because the opinions of two discriminators will neither be perfectly correlated nor are they completely independent.Given these previous difficulties, I was surprised to see that the present paper can give simple expressions for the players' payoffs (Equations 5-9 in the Materials and methods section), which also hold for E=0. Unfortunately Equations 5-9 are not derived in sufficient detail, which makes it impossible to check whether they are correct.In these formulas, g is the frequency of good individuals within the population, whereas the authors seem to use g^2^ for the probability that two individuals both assign a good reputation to a random recipient. This seems to presume complete independence of individual opinions.I would like to ask the authors to provide a more detailed derivation of the payoff Equations 5-9. In particular, the authors should make clear how they circumvented the problems described in the above mentioned paper of Okada et al., 2018.The same comment also applies to the formulas in the Materials and methods section on the evolution of empathy.

The reviewer is correct: our initial expressions involving *g*^2^ for the probability that two individuals see a focal player as “good” assumed complete independence of evaluations. This assumption was wrong, and we have corrected it in the revision (including changes in the Results and Materials and methods sections, Figure 2, Figure 2—figure supplement 1, and Figure 5—figure supplement 1). In fact, our model assumes that every individual interacts with all recipients, and then an observer makes her moral evaluation of a donor on the basis a single, randomly chosen interaction. In an infinite population each interaction is therefore observed only once, as in “solitary observation” approach of Okada et al., 2018. We have re-written Equations 5-9 in terms of the probability that two individuals assess the same donor as good (*g*_2_ = *f_X_g_X_*^2^ +*f_Y_ g_Y_*^2^ +*f_Z_g_Z_*^2^), bad (*b*_2_ = *f_X_*(1−*g_X_*)^2^ +*f_Y_*(1−*g_Y_*)^2^ +*f_Z_*(1−*g_Z_*)^2^), or disagree about his reputation *d*_2_ = 1*/*2(1−*g*_2_−*b*_2_). We have updated equations (1) and (2) accordingly. (This issue does not apply to the Equations 10-12, as those assume that the population consists solely of discriminators.)

Our qualitative and even quantitative results remain largely unchanged by these corrections. Nonetheless we are very grateful to the reviewer for noticing the erroneous independence approximation, because correcting this error has clarified our model and its connection to the recent literature.

2) The paper is based on the assumption that all players in the population employ the same social norm. As the authors note in the last paragraph of their discussion, it is not clear whether cooperation remains stable if mutants were allowed to switch to a self-serving social norm (by assigning a bad reputation to everyone, and then defect against everyone). It appears that such a mutant is a major threat to all social norms studied herein, especially when the empathy probability is high.I completely agree with the authors that a full analysis of such mutations is beyond the scope of the present paper. However, I would appreciate if the authors could give some basic argument why their results can be expected to be robust.As one possibility, the authors could repeat the analysis in Figure 2 or Figure 3. But instead of assuming that ALLD and ALLC players apply the same social norm as the discriminators, they could assume that ALLD players always assign a bad reputation to everyone, whereas ALLC players always assign a good reputation to everyone. This would correspond to a case in which the social norms applied by the unconditional types are consistent with their actions.If such an analysis turns out to be infeasible, I suggest the authors explain under which circumstances we can reasonably assume all members of a population to use the same social norm.

The full analysis of strategy-norm co-evolution under high empathy is part of our future work in this area. However, here we would like to point out that a selfish norm that assigns bad reputations to everyone together with ALLD strategy might not necessarily invade, since other norms can still punish defection against ‘bad’ while retaining rewards for cooperation with the ‘good’. In other words, while in the fully empathetic case the observer takes into account the recipient’s reputation from the donor’s perspective, the norm used to make the moral evaluation still belongs to the observer and not the donor. We discuss this point briefly in our conclusion section, but otherwise restrict our analysis to a shared norm (which is the standard assumption in the literature, presumably on the grounds that norm deviation is often punished, in practice).

3) The results in Figure 3C are either wrong, or they require more explanation.In this figure, the authors show that for a rather small mutation rate (μ=0.01), almost all social norms lead to similar cooperation rates and that cooperation rates are largely independent of the empathy parameter. There are two reasons why Figure 3C is somewhat odd:i) It suggests that empathy can only help for extremely small mutation rates.ii) It suggests that in the case of no empathy E=0, Stern Judging can yield roughly 40% cooperation. This is somewhat surprising, given that previous papers have found that under private information, a Stern Judging norm typically leads to an ALLD population with almost no cooperation at all (e.g. Okada, Sasaki and Nakai, 2017; Hilbe et al., 2018).

In Figure 3C the mutation rate is *µ* = 0.01 with population size *N* = 100, so that *θ* = 1. This is the highest mutation rate that we consider, and it corresponds to the case where stochastic effects dominate, with the population moving freely between the cooperative and non-cooperative basins of attraction. All norms therefore lead to the nearly identical mean rates of cooperation. (The same result in seen in Figure 2 from Santos et al., 2016.) Indeed, empathy has no effect when stochastic exploration dominates the strategy dynamics, and this is an intuitive result. For Stern Judging our findings agree well with the previously published results (with *E* = 0) under lower mutation rates. By comparison, the results described in Hilbe et al., 2018, were derived in the limit of *µ* → 0, which is a qualitative different regime compared to *θ* = 1. We are not surprised to find different results under *θ* = 1 compared to *µ* → 0, and indeed our results for *θ* = 1 coincide with previous studies in this regime of mutation.

On a similar note, I would appreciate if the authors could be more clear in the main text about the relevant timescales. From reading the Materials and methods section, it seems to me that the authors assume that in the infinite population model, reputations are assumed to reach a steady state before evolutionary updating occurs. In the finite population simulations, evolutionary updating may occur before the players' reputations have reached a steady state, is this correct?

Thank you for this suggestion. In the infinite-population analysis we indeed assume that reputation frequencies reach their steady state before the payoffs are updated and before selection takes place. In our Monte Carlo simulations each individual plays *N* games before reputations are updated; the evolutionary updating based on accumulated payoffs follows the moral-evaluation step. It is therefore possible that reputations do not reach steady state before the evolutionary updating, although for weak selection and close to the strategy equilibrium the analysis of the infinite-population model provides a very good approximation of the behavior observed in the finite-population simulations. We have clarified this issue of timescales in the main text (first paragraph of Results section).